# Availability, prices and affordability of essential medicines in Zhejiang Province, China

**Zuojun Dong**, **Qiucheng Tao, Bobo Yan, Guojun Sun**\*

College of Pharmacy, Zhejiang University of Technology, Hangzhou, Zhejiang, China

\* zmsgj@zjut.edu.cn

## Abstract

### Objective

To evaluate the availability, prices, and affordability of essential medicines in Zhejiang Province, China.

### Methods

The survey was carried out in Zhejiang Province in 2018 following the methodology of the World Health Organization (WHO) and Health Action International (HAI). This method is an international standard method.Data on 50 medicines were collected from public health facilities and private pharmacies. Medication prices were compared with international reference prices to obtain a median price ratio. The affordability of medicines was measured based on the daily wage of the lowest-paid unskilled government worker. In private pharmacies, the mean availability of Originator Brands (OBs) and Lowest-priced Generics (LPGs) was 36.7% and 40.3%, respectively.

### Findings

The effects of the mean availability of OBs and LPGs were seen in private pharmacies. Correspondingly, the average availability of OBs and LPGs was 41.8% and 35.1% in the public sector, respectively. In the public sector, the median price ratios (MPRs) were 5.21 for generics and 13.49 for OBs. In the private sector, the MPRs were 4.94 for generics and 14.75 for OBs. Treating common diseases with LPGs was generally affordable, while treatment with OBs was less affordable.

### Conclusions

In Zhejiang Province, low availability was observed for medicines surveyed in the public and private sectors. Price differences between originator brands and generics in both sectors are apparent. OBs were more expensive than LPGs in both the public and private sectors. Low availability affects access to essential medicines. Policy measures should be taken to improve the availability of essential medicines.

**Data Availability Statement:** All relevant data are contained within the manuscript and its Supporting Information files.

**Funding:** The authors received no specific funding for this work.

**Competing interests:** The authors have declared that no competing interests exist.

## Introduction

Medicines, especially essential medicines, play a significant role in health care [1]. Though health care is a basic human right [2], this fundamental right cannot be met without fair access to essential medicines. Essential medicines satisfy the health care needs of the population [3]. They are selected based on disease prevalence, evidence of efficacy and safety, and comparative cost-effectiveness [4]. In China, the national essential medicine system was implemented in 2009 [5].

In Zhejiang, all medicines stocked by public health care institutions are procured by the Bureau of Drug Price Bidding, Zhejiang Provincial Department of Health. The procurement complies with the requirement of the province's unification. All the medicines were distributed with zero mark-up [6].

Zhejiang, located in eastern China, has a population of 56.57 million and has 11 cities. It is among the top-five provinces in China in terms of economic development. In Zhejiang, total expenditure on health care in 2019 was 2.11% of gross domestic product (GDP), just under the 5% level recommended by the World Health Organization (WHO). Meanwhile, in recent years, some essential medicines with low prices and high quality are short in supply. For instance, in 2017, there was an acute shortage of mercaptopurine tablets. The present study aimed to examine the availability of essential drugs. Few surveys have evaluated the availability and affordability of medicines in Zhejiang Province, China. This was the context in which we conducted this study. This survey can help us to understand the availability and affordability of essential medicines in Zhejiang Province.

## Survey areas

For the price and availability survey, Hangzhou, the capital city of Zhejiang Province, was chosen as the major urban center, and an additional five areas (Ningbo, Shaoxing, Huzhou, Taizhou, and Jiaxing) within one day of travel of Hangzhou were randomly selected. We investigated price components in two areas: Hangzhou (the main urban center) and Taizhou (far from the urban center) to collect the price markups in the supply chain.

### Data collection and entry

Six well-trained data collectors visited medicine outlets and collected data on medicine prices (procurement price and patient price in public sector outlets, and patient price in private sector outlets) and availability, using standardized price collection forms. No specific permits were required for the described field studies. Three trained personnel entered the data into the standardized WHO/HAI Excel Workbook using a double-entry technique. Five data collectors visited key informants along the drug supply chain and collected price component data. Then, we entered this information into the WHO/HAI Excel workbook.

### Ethics statement

The study was approved by the Ethics Committee of People's Government of Zhejiang Province, and Zhejiang Food and Drug Administration approved the study before data collection. The participants were informed of the aims of our study prior to participation. All participants provided signed written informed consent forms.

### Statistical analysis

Using the standardized WHO/HAI methodology, we conducted a study on the availability, prices, and affordability of 50 medicines in Zhejiang Province, China [7]. Among the

medicines selected in this study, 12 belonged to the global core list medicines recommended by WHO/HAI, and the other 38 were selected as supplementary drugs. The supplementary medicines were determined according to the local disease burden, the 2012 National Essential Medicine List (NEML) [8], local needs, and the opinions of experts. Other factors were also taken into account; for instance, each medicine selected for the supplementary list had to have an international reference price. During the study, we investigated and collected the prices of Originator Brands (OBs) and Lowest-priced Generics (LPGs) for 50 drugs and then compared these prices with international reference prices (IRPs). OBs are products of the same specification produced by the originator manufacturer of the drug, and LPGs are the cheapest generic drugs found by the University research institute. This study adopted the standard sampling survey method provided by the WHO/HAI to determine the cheapest LPGs. Our research focuses on sample surveys, and surveys are conducted in the form of questionnaires. In the hospitals and pharmacies surveyed, if there is no OB, a generic drug with the same name and dosage form will be used as an LPG for research. If there is no generic drug with the same name and dosage form, there is no universal substitute, and it will be displayed as "0" in the table.

All data for this study were collected from March to May 2018. Data collection included six cities in Zhejiang Province. In each area surveyed, five public pharmacies were selected from the list provided by the Health and Family Planning Commission of Zhejiang: one main hospital, two secondary hospitals, and two primary hospitals by convenience sampling. Private pharmacy outlets were randomly selected based on their active registration status and scale [7]. The private pharmacy sample included 30 pharmacy outlets. The availability of individual medicines is calculated as the percentage (%) of medicine outlets where the medicine was found. The availability of medicines was recorded on the day of data collection.

To facilitate comparisons among countries, medicine prices obtained during the survey are expressed as median price ratios (MPRs), referring to the ratio of a medicine's median unit price across outlets to the median unit price in the Management Sciences for Health 2015 Price Indicator Guide [9]. The Management Sciences for Health 2015 Price Indicator Guide was chosen because of the frequency of its updates, ease of access, and relative stability.

On the basis of the standard WHO/HAI methodology [7], the affordability of core medicines for common diseases in Zhejiang Province was assessed. The affordability was calculated by comparing the total cost of medicines prescribed at a standard dose with the daily wage of the lowest-paid unskilled government worker, which was RMB 68.9655 per day at the time of the survey, based on figures from Zhejiang Province Human Resources and Social Security Department [10].

## Results

### Availability

In public pharmacies, the mean availability of OBs and LPGs was 41.8% and 35.1%, respectively. In the private pharmacies, the average availability of OBs and LPGs was 36.7% and 40.3%, respectively. Separate availability analysis of 38 medicines listed on the NEML indicated that the mean availability in private sector retail pharmacies was 35.0% for OBs and 39.5% for LPGs, while average availability in the public sector was 41.4% for OBs and 35.4% for LPGs.

We refer to Table 1, Table 1 shows the availability of individual medicines in the public and private sectors. As shown in the table, the availability of the selected medicines in both public and private pharmacies was low. Among the OBs, amlodipine and atorvastatin had the highest availability in both sectors.

**Table 1. Availability of individual medicines in public pharmacies and private pharmacies.**

| Name of medicine | Public pharmacies | | Private pharmacies | |
|---|---|---|---|---|
| | OBs availability (%) | LPGs availability (%) | OBs availability (%) | LPGs availability (%) |
| Albendazole | 66.7 | 0 | 96.7 | 40.0 |
| Amitrityline | 0 | 46.7 | 0 | 23.3 |
| Amlodipine | 93.3 | 63.3 | 80.0 | 96.7 |
| Amoxicillin | 0 | 43.3 | 0 | 63.3 |
| Atenolol | 0 | 0 | 0 | 0 |
| Atorvastatin | 93.3 | 43.3 | 80.0 | 60.0 |
| Azithromycin | 26.7 | 66.7 | 20.0 | 63.3 |
| Bisoprolol | 66.7 | 40.0 | 50.0 | 16.7 |
| Captopril | 0 | 56.7 | 6.7 | 63.3 |
| Cefalexin | 0 | 10.0 | 0 | 13.3 |
| Ceftriaxone injection | 66.7 | 6.7 | 0 | 0 |
| Cefuroxime | 30.0 | 40.0 | 13.3 | 33.3 |
| Cetirizine | 23.3 | 26.7 | 20.0 | 80.0 |
| Chlorphenamine Maleate | 0 | 56.7 | 0 | 80.0 |
| Ciprofloxacin | 0 | 6.7 | 0 | 10.0 |
| Clarithromycin | 6.7 | 30.0 | 3.3 | 60.0 |
| Clomipramine | 3.3 | 16.7 | 0 | 3.3 |
| Clopidogrel | 93.3 | 26.7 | 53.3 | 26.7 |
| Co-trimoxazole suspension | 0 | 0 | 0 | 3.3 |
| Diazepam | 0 | 3.0 | 0 | 0 |
| Diclofenac Sodium | 23.3 | 6.7 | 73.3 | 13.3 |
| Digoxin | 0 | 86.7 | 0 | 43.3 |
| Diphenhydramine | 0 | 3.3 | 0 | 3.3 |
| Doxycycline | 0 | 46.7 | 0 | 16.7 |
| Enalapril | 3.3 | 26.7 | 3.3 | 63.3 |
| Erytromycin | 0 | 23.3 | 0 | 3.3 |
| Gliclazide | 3.3 | 16.7 | 40.0 | 73.3 |
| Glimepiride | 70.0 | 56.7 | 56.7 | 83.3 |
| Hydrochlorothiazide | 0 | 86.7 | 0 | 40.0 |
| Ibuprofen | 0 | 20.0 | 0 | 36.7 |
| Irbesartan | 86.7 | 66.7 | 80.0 | 93.3 |
| Levofloxacin | 0 | 53.3 | 0 | 26.7 |
| Lisinopril | 0 | 3.3 | 0 | 10.0 |
| Lortadine | 43.3 | 20.0 | 80.0 | 96.7 |
| Losartan | 40.0 | 66.7 | 76.7 | 63.3 |
| Mebendazole | 23.3 | 0 | 13.3 | 3.3 |
| Metformin | 66.7 | 56.7 | 63.3 | 40.0 |
| Metronidazle | 0 | 73.3 | 0 | 76.7 |
| Mupriocin | 50.0 | 40.0 | 73.3 | 60.0 |
| Nifedipine | 0 | 53.3 | 0 | 86.7 |
| Nimodipine | 53.3 | 6.7 | 16.7 | 6.7 |
| Omeprazole | 20.0 | 73.3 | 43.3 | 86.7 |
| Oseltamovir | 60.0 | 53.3 | 3.3 | 0 |
| Paracetamol | 23.3 | 23.3 | 3.3 | 56.7 |
| Promethazine HCL | 0 | 13.3 | 0 | 6.7 |
| Propranolol | 0 | 46.7 | 0 | 33.3 |

*(Continued)*

**Table 1.** (Continued)

| Name of medicine | Public pharmacies | | Private pharmacies | |
|---|---|---|---|---|
| | OBs availability (%) | LPGs availability (%) | OBs availability (%) | LPGs availability (%) |
| Salbutamol inhaler | 70.0 | 0 | 40.0 | 50.0 |
| Sertraline | 46.7 | 46.7 | 16.7 | 6.7 |
| Simvastatin | 43.3 | 53.3 | 66.7 | 53.3 |
| Tinidazole | 0 | 46.7 | 0 | 46.7 |

Note: 0 means the drug was not obtained in the pharmacy.

We refer to Table 2, Table 2 lists the availability of medicines in the public pharmacies and the private pharmacies. Twenty-eight OBs were found in the public sector and 23 in the private sector, while 45 LPGs were found in the public sector and 46 in the private sector. In the public sector, only 13 OBs and 16 LPGs had >50% availability. In the private sector, only 13 OBs and 21 LPGs had>50% availability.

## Price

**Prices of the public pharmacies.** We refer to Table 3.

The results in Table 3 indicate that generic products in the public sector were generally sold at more than five times the international reference prices. Half of these were sold from 2.77 times (25th percentile) to 8.44 times (75th percentile) more than their reference prices. There was a large difference between the minimum MPR of 0.15 (captopril) and the maximum MPR of 43.89 (omeprazole).

**Price of private pharmacies.** We refer to Table 4 and Fig 1, The results in Table 4 show that prices in the private sector are much higher than the international reference prices. Of the originator brand products, omeprazole was found to be the highest priced at over 130 times the international reference price. Four originator brand products are over 40 times more expensive than their international reference prices. They also have the highest-priced generic products. The diagram below illustrates the MPRs of these four medicines in the private sector.

We refer to Table 5, The results in Table 5 show that the patient price in the public sector is 13.41 for OBs, lower than that in the private sector while for LPGs it is 5.21, higher than that in the private sector.

## Affordability

The affordability was calculated using the median prices collected during the survey [7]. Table 5 shows the affordability associated with the seven common diseases. OB products were

**Table 2. Availability of medicines in the public sector and the private sector.**

| Availability | Public facilities | | Private pharmacies | |
|---|---|---|---|---|
| | Originator brand | Lowest-priced generic | Originator brand | Lowest-priced generic |
| Medicines not found in any outlets | 22 | 5 | 23 | 4 |
| Medicines found in fewer than 25% of outlets | 9 | 15 | 11 | 15 |
| Medicines found in 25 to 50% of outlets | 6 | 14 | 3 | 10 |
| Medicines found in 50 to 75% of outlets | 9 | 14 | 7 | 12 |
| Medicines found in more than 75% of outlets | 4 | 2 | 6 | 9 |

N = 50.

**Table 3.  Prices in the public sector compared to international reference prices.**

|  | Lowest-priced Generic (times) |
|---|---|
| MPR | 5.21 |
| 25th percentile MPR | 2.77 |
| 75th percentile MPR | 8.44 |
| Minimum MPR | 0.15 |
| Maximum MPR | 43.89 |

generally less affordable than the lowest-priced medicines in both the public and private sectors. However, in the private sector, some treatment costs were high. For instance, treating peptic ulcer with omeprazole required 5.2 days' wages. The treatment course of omeprazole is 4–8 weeks.

We refer to Table 6.

## Discussion

Some studies on access to essential medicines have been conducted in China using the WHO/HAI methodology [11–13]. The present study is the first to apply this methodology to the availability of essential medicines in Zhejiang Province in China. The findings of this study offer a report on the availability, prices, and affordability of essential medicines in Zhejiang Province in China. We conducted this survey in six cities in Zhejiang Province. The results suggest that the availability of 50 essential medicines is low in the public and private sectors. In private pharmacies, the mean availability of OBs and LPGs was 36.7% and 40.3%, respectively. The average availability of OBs and LPGs was 41.8% and 35.1% in the public sector. Compared with the study of Minghuan Jiang et al. [12], our findings showed a higher availability of essential medicines in both the public and private sectors. Their study showed that, in Shaanxi Province, the mean availability of OBs and LPGs was 12.9% and 29.5% in private pharmacies, respectively [11]. The mean availability of OBs and LPGs was 7.1% and 20.0% in public hospitals.

On January 17, 2019, the Chinese government issued the "National Organized Drug Centralized Procurement and Use Pilot Program," marking the beginning of the Chinese government's formal implementation of the centralized drug procurement policy, which has greatly reduced the prices of drugs in China. This policy is labelled the "4+7" centralized procurement policy. The Chinese government forces bid-winning companies to lower the price of medicines. The bid-winning medicines of bid-winning companies can gain more market share leading to lower medicine prices through large purchases.

There are three reasons for these findings. First, the low profit margins of essential drugs have prompted the public and private sectors to support other high-profit equivalent

**Table 4.  Prices in the private sector compared to international reference prices.**

|  | Originator Brand | Lowest-priced Generic |
|---|---|---|
| MPR | 14.75 | 4.94 |
| 25th percentile MPR | 7.67 | 2.02 |
| 75th percentile MPR | 25.12 | 9.30 |
| Minimum MPR | 1.91 | 0.19 |
| Maximum MPR | 134.45 | 28.75 |
| Number of medicines included | 22 | 36 |

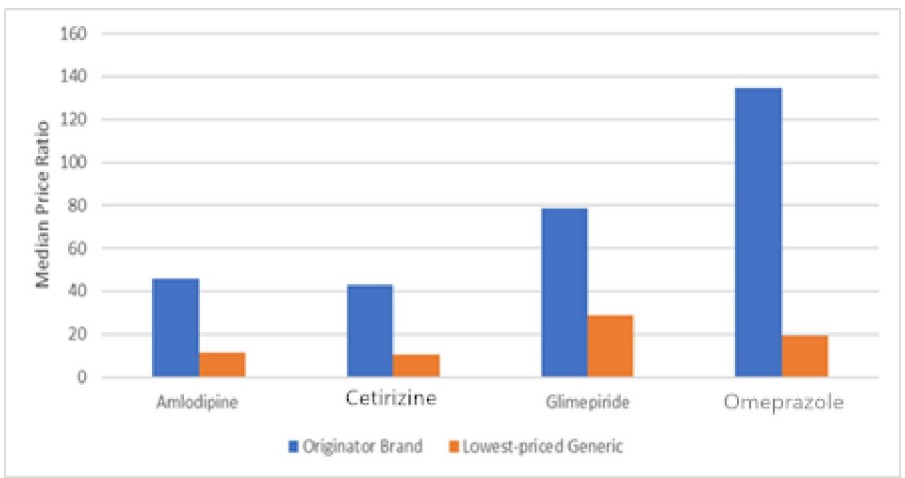

**Fig 1. MPRs of these four medicines in the private sector.**

alternative drug treatment schemes. In Zhejiang Province, public medical institutions adopt a zero-margin drug sales policy; that is, the purchase price is the same as the price patients pay [14]. In the past, medical institutions could increase the price of drugs by 15% on the basis of purchase prices. Because of the lack of sales profits, medical institutions are more inclined to seek profits in equipment inspection or operation, and the willingness to increase the supply of drugs is significantly reduced. Second, the government's essential drug policy prohibits the sale of essential drugs by medical institutions at premium prices, and by allocating funds to hospitals to maintain their operations, government subsidies usually only account for 10% of operating expenses [15]. It can be said that hospital pharmacies are now the pure-cost expenditure units of medical institutions, so they tend to reduce the scale of drug supply. In China, public pharmacies do not charge for prescription reviews and also do not charge preparation service fees for drugs sold to patients. Their profit comes from the premium between the purchase and sale of medicines, which is very different from how pharmacies in the United States and other countries operate.

Although some hospitals have charged pharmaceutical services because pharmaceutical services in Zhejiang or China are still in their infancy, general hospitals do not have the ability to provide better pharmaceutical services, so only a few hospitals in individual areas charge pharmacist service fees [16]. The essential drug list is an important guideline for the implementation of an essential drug system. China has implemented an essential drug system for several years. Primary medical institutions have achieved full coverage of essential drugs. With the progress of medical and health system reforms, the next goal of the essential drug system is inevitable. It is to increase the use rate of essential medicines in secondary and tertiary hospitals. Third, most essential medicines are commonly used medicines, which are the main target of price reduction in recent years. The profit margin is relatively small, which makes enterprises unenthusiastic about maximizing profits. The "4+7 bulk quantity purchasing policy"

**Table 5. MPRs for medicines found in both the public and private pharmacies.**

| Product type | MPR public sector patient prices | MPR private sector patient prices | % difference private to public |
|---|---|---|---|
| OBs (21) | 13.41 | 14.85 | 10.7% |
| LPGs (32) | 5.21 | 4.94 | -5.3% |

**Table 6. Affordability of core medicines for common diseases in public sector and private sector.**

| Disease Condition | Drug name | Product Type | No. of units a day | Duration (days) | Day's wage (Public pharmacies) | Day's wage (Private pharmacies) |
|---|---|---|---|---|---|---|
| Asthma | salbutamol | Originator | 200 | as needed | 0.3 | 0.3 |
| | | Lowest-priced | 200 | as needed | NA | 0.3 |
| Diabetes | Metformin | Originator | 3 | 30 | 1.5 | 1.6 |
| | | Lowest-priced | 3 | 30 | 0.6 | 0.8 |
| Hypertension | Captopril | Originator | 2 | 30 | NA | NA |
| | | Lowest-Priced | 2 | 30 | NA | NA |
| | Bisoprolol | Originator | 2 | 30 | 2.5 | 2.5 |
| | | Lowest-Priced | 2 | 30 | 1.8 | 1.8 |
| Hypercholesterolaemia | simvastatin | Originator | 1 | 30 | 1.3 | 1.3 |
| | | Lowest-Priced | 1 | 30 | 0.4 | 0.8 |
| Depression | amitriptyline | Originator | 3 | 30 | NA | NA |
| | | Lowest-Priced | 3 | 30 | 0.2 | 0.2 |
| Adult respiratory infection | amoxicillin | Originator | 3 | 7 | NA | NA |
| | | Lowest-Priced | 3 | 7 | 0.2 | 0.3 |
| | ciprofloxacin | Originator | 2 | 7 | NA | NA |
| | | Lowest-Priced | 2 | 7 | NA | NA |
| | ceftriaxone | Originator | 1 | 1 | 0.7 | NA |
| | | Lowest-Priced | 1 | 1 | NA | NA |
| Ulcer | omeprazole | Originator | 1 | 30 | 4.6 | 5.2 |
| | | Lowest-Priced | 1 | 30 | 1.7 | 0.8 |

Note: NA, not available.

Duration: Duration of treatment course over a one-month period.

means that, with the approval of the Central Committee for Comprehensive and Deepening Reform, the state organizes a pilot project of centralized drug purchasing, covering 11 cities in Beijing, Tianjin, Shanghai, Chongqing and Shenyang, Dalian, Xiamen, Guangzhou, Shenzhen, Chengdu, and Xi'an. If we take the "4 + 7" bidding purchase as an example, average reduction of more than 50%, the highest drop in drugs reached 92%. In this year's bidding and procurement, the "4 + 7" model is also being followed up in areas outside the pilot project. It is generally believed that the prices of essential drugs will continue to fall. Prices have declined since the "4 + 7" policy, which reduces the burden on patients, but the shortage of drugs is also increasing. For example, the Chinese government has established a drug shortage–monitoring mechanism [17], and provinces have issued their own drug shortage catalogs [18].

The "4+7" policy is a national policy, but the provinces will make appropriate adjustments according to their actual conditions to better realize the availability and affordability of essential medicines [19]. For example, on the December 6, 2019, the National Medical Security Administration of China issued the "Opinions on Doing a Good Job in Current Drug Price Management." Article 1 of Chapter 3 states that in case of drug shortages, operators are allowed to quote their prices directly online. The medical institution purchases at the online price or further negotiates with the operator for purchase. On June 28, 2020, the Zhejiang

provincial government issued the "Zhejiang Province to Enhance the Function of the Centralized Pharmaceutical Procurement Platform to Promote the Reform of the Full Coverage of Medical Insurance and Drug Payment Standards." Chapter III noted that for clinical needs, public medical institutions are entitled to fair negotiations with enterprises to determine the purchase price within the limit of 1% of the settlement amount of drugs in the previous year and purchase products themselves through the provincial drug equipment purchase platform, if there are no online traded products within the platform or they cannot be purchased at the set price within the platform [20].

This study has several limitations. First, we use the research method based on WHO/HAI, which may diverge from the real situation in Zhejiang Province. A better method may need to be developed in the future. Second, the relevant data are only applicable to the date of data collection and may not reflect the average availability over time. Finally, this study analyzes the price components of essential medicines in China using the standardized WHO/HAI method. However, only some medicines were included in the analysis. For confidentiality reasons, some data have errors and are missing.

## Conclusion

In Zhejiang Province, the supply of drugs surveyed by the public and private sectors is inadequate. The price difference between the original brands and generic drugs in the two industries is obvious. In the public and private sectors, OBs are more expensive than LPGs. Relevant measures should be taken to improve the supply of essential drugs.

First, the government should increase subsidies for health insurance, investment in this field has been increasing in recent years. For example, in 2019, in Zhejiang Province, the basic medical insurance income is each worker's basic medical insurance allotment, the income of the basic medical insurance fund for Chinese workers and urban and rural residents increased by 9.94% year on year, and the income of the basic medical insurance fund for urban and rural residents increased by 7.71% year on year. Second, the government should improve the availability and affordability of essential medicines in public pharmacies. Now, some of the measures, such as lowering the barriers for pharmacies to enter the health care system and allowing them to sell essential drugs at higher prices than medical institutions, should be effective. Third, allowing pharmacists in public pharmacies to charge pharmaceutical service fees will significantly improve the income structure of public pharmacies and encourage pharmacists to provide more types of drugs and better pharmaceutical care, instead of paying attention to drugs with high premium prices. The good news is that the service fees pharmacists charge for drug preparation may be recognized by law. Article 5 of the second draft of the pharmacist law of the People's Republic of China, adopted by the Chinese government on June 18, 2020, required pharmacists to provide pharmaceutical professional technical services to patients, which indicated that pharmacists' service charge is likely to become a reality in the future.

In this study, we studied the affordability of medicines for seven common conditions. As shown, most LPGs for these common diseases were affordable in both sectors. The improvement in people's living standards enables them to afford high medical expenses.

## Supporting information

**S1 File.**
(ZIP)

## Author Contributions

**Conceptualization:** Guojun Sun.

**Data curation:** Bobo Yan.

**Formal analysis:** Qiucheng Tao.

**Funding acquisition:** Bobo Yan.

**Investigation:** Guojun Sun.

**Project administration:** Zuojun Dong.

**Resources:** Qiucheng Tao.

**Writing – review & editing:** Zuojun Dong.

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
