## [Decision Letter · Decision Letter 0]

14 May 2020

PONE-D-19-30034

Availability, prices and affordability of essential medicines in Zhejiang Province, China

PLOS ONE

Dear Dong,

Thank you for submitting your manuscript to PLOS ONE. After careful consideration, we feel that it has merit but does not fully meet PLOS ONE’s publication criteria as it currently stands. Therefore, we invite you to submit a revised version of the manuscript that addresses the points raised during the review process.

We would appreciate receiving your revised manuscript by Jun 28 2020 11:59PM. To enhance the reproducibility of your results, we recommend that if applicable you deposit your laboratory protocols in protocols.io, where a protocol can be assigned its own identifier (DOI) such that it can be cited independently in the future. For instructions see: http://journals.plos.org/plosone/s/submission-guidelines#loc-laboratory-protocols

We look forward to receiving your revised manuscript.

Kind regards,

Susan Horton

Academic Editor

PLOS ONE

Journal Requirements:

2.  Please address the following:

- Please ensure you have thoroughly discussed any potential limitations of this study within the Discussion section.

- Please ensure your methods have been described in sufficient detail so that these analyses could be performed again. In the text you state "Data collection also uses the WHO/HAI methodology", which is not sufficient.

- Please include any sample size calculations performed prior to data analysis. If this has not been performed please justidy the reasons. Please refer to our statistical reporting guidelines for assistance (https://journals.plos.org/plosone/s/submission-guidelines.#loc-statistical-reporting).

Thank you for your attention to these queries.

https://journals.plos.org/plosone/article?id=10.1371/journal.pone.0070836

In your revision ensure you cite all your sources (including your own works), and quote or rephrase any duplicated text outside the methods section. Further consideration is dependent on these concerns being addressed.

'NO - Include this sentence at the end of your statement: The funders had no role in study design, data collection and analysis, decision to publish, or preparation of the manuscript.'

6. Please amend the manuscript submission data (via Edit Submission) to include author Bobo Yan.

7. Please amend either the abstract on the online submission form (via Edit Submission) or the abstract in the manuscript so that they are identical.

8. Please ensure that you refer to Figure 1 in your text as, if accepted, production will need this reference to link the reader to the figure.

9. We note you have included tables to which you do not refer in the text of your manuscript. Please ensure that you refer to Tables 2, 5 and 7 in your text; if accepted, production will need this reference to link the reader to the Tables.

10. Please include captions for your Supporting Information files at the end of your manuscript, and update any in-text citations to match accordingly. Please see our Supporting Information guidelines for more information: http://journals.plos.org/plosone/s/supporting-information.

Additional Editor Comments:

I have recently been invited to be a Guest Editor for this manuscript, to help move it along in the review process. I see that you have been waiting a long time for the reviews, so I conducted my own review (below, as Reviewer 2) to add to that of another reviewer (at end of the letter). Please address both sets of comments.

Sincerely, Sue Horton (University Research Chair in Global Health Economics, University of Waterloo)

Reviewer 2 comments

Availability and affordability of medicines is an interesting topic. This paper repeats an identical format and analysis from two other provinces of China (Shaanxi and Jiangsu) in a third province (Zhejiang), which in turn use a 2003 methodology developed by WHO and Health Action International. which was applied in 50 countries studies by 2008. Subsequent to the publication of the comparative results from the WHO study, the methodology has also been used to analyze the same parameters for pediatric drugs, diabetes drugs, cardiovascular drugs, drugs for noncommunicable diseases etc in multiple countries (Nepal, Nigeria, Ethiopia, Anhui province of China, Bangladesh, among many others). Given that China has 32 provinces/municipalities/autonomous regions, there is obviously a lot of scope for many such surveys to be conducted in China. This paper does innovate slightly by relating the results to the “4+7” policy although the way this is presented is a bit difficult for the reader to follow.

There are some methodological issues that would be important to address:

1. It’s not clear why Tables 1 and 2 are presented separately, since the availabilities in Table 3 and the price summaries in Tables 4 and 5 appear to be based on the combined data.

2. Tables 1 and 2 at times use “0” and other times “-“ in the columns. Initially I assumed that “-“ meant not applicable and that these items were excluded from Table 4 but that appears not to be the case, and probably “0” and “-“ actually mean the same thing.

3. Table 4 has a note below the table stating this applies to 37 medicines (although the superscript “a” does not appear in the table, so I am guessing here). There is no reason given for this, since there are more than 37 LPG drugs available in that sector according to Tables 1 and 2.

4. Similarly, in Table 5 has a line in the table this time indicating that the comparisons 22 and 36 medicines, however this does not seem to equate to the number of cases where the drug was available in the private sector. Some explanation would be desirable.

5. I have an overall concern with the WHO-HAI methodology. The importance of non-availability of the “lowest priced generic” requires careful definition. Is the “lowest priced generic” defined for China (or for Zhejiang province) and if so how? How much difference is there between the prices of the lowest priced generic and the second lowest priced generic for these drugs? What is the policy significance of the lowest-priced one being unavailable in a circumstance where possibility the second lowest-priced one is available, and how does that compare to a situation where the lowest-priced one is considerably less expensive than the next lowest-price one? It is possible that when the WHO originated the methodology back in 2003 fewer generics were available, so this wasn’t as much of an issue as it perhaps is now.

6. How did the authors take account of the fact that the international reference prices were for 2015, while the data were collected in 2018. Do international pharmaceutical prices not change much over three years?

7. Given that the policy issues are the main area in which this paper distinguishes itself from scores (or hundreds) of similar studies, it would be worth getting an English-first-language health expert to read the discussion and conclusion sections and make the policy implications clearer for example the term “subsidies” in the Discussion section sometimes seems to mean government contributions to the hospital budget rather than a subsidy per se; and “The essence of “4 + 7” is to exchange price by quantity” would be better explained “…is to use bulk purchases to drive down prices” or something similar.

Reviewers' comments:

Reviewer's Responses to Questions

**Comments to the Author**

1. Is the manuscript technically sound, and do the data support the conclusions?

Reviewer #1: Yes

2. Has the statistical analysis been performed appropriately and rigorously? 

Reviewer #1: Yes

3. Have the authors made all data underlying the findings in their manuscript fully available?

Reviewer #1: Yes

4. Is the manuscript presented in an intelligible fashion and written in standard English?

Reviewer #1: No

5. Review Comments to the Author

Reviewer #1: Dear Authors,

The topic on medicines access is still relevant and important in many ways. It is good that such studies are still been carried out. However, this article need improvements:

1. Overall the article need language improvement by an expert; thus I will not comment on the matters related to language and grammar.

2. Abstract - Conclusions - affordability is one of the main objs but it was not mentioned.

3. Methods section - you mentioned 'an additional 5 cities' but you only mentioned 4+1 - it is confusing - 5 or 6 cities in total?

4. Methods section - what does 'scale' mean in this context ?

5. many researchers used the WHO/HAI methodology but why this methodology? any strengths/advantages? please explain.

6. Figure 1 - check the spelling for Omeprazole; and the use of cap vs small letter

7. Table 7 - font type - be consistent with the main text; check the use of cap vs small letters for drug name

8. Discussion - does this province use diff policy and regulation from other provinces in China? good for the International readers to understand

9. Discussion - lack of studies from other countries that are relevant; were not included in the discussion

10. Conclusion - sentence 1 - 'inadequate' - what does this clearly mean? shortages? not available?

11. Include recommendations section at the end of the Discussion section - move the ones mentioned in the Conclusion section.

12. Include the strength, limitations and study implications in the last part of the Discussion section

Reviewer #1: Yes: Professor Mohamed Izham Mohamed Ibrahim, PhD

---

## [Author Response · Author response to Decision Letter 0]

10 Jul 2020

Dear Susan Horton

Thanks you very much for your comments and suggestions. 

As suggested, We have made amendments to the academic papers, in accordance with the requirements of the academic editors, and responded to the comments made by the reviewers. The details are attached below. Some of our basic research data is written in Chinese, which may be difficult to understand.

We have polished English and revised academic papers.We have revised the manuscript, according to the comments and suggestions of reviewers and editor, and responded, point by point to, the comments as listed below. Some of the references we cited are published in domestic academic journals in China and may not be found in foreign databases.

I would like to re-submit this revised manuscript to PLOS ONE, and hope it is acceptable for publication in the journal.

Looking forward to hearing from you soon.

With kindest regards,

Yours Sincerely

QiuCheng Tao

---

## [Decision Letter · Decision Letter 1]

28 Jul 2020

PONE-D-19-30034R1

Availability, prices and affordability of essential medicines in Zhejiang Province, 

China

PLOS ONE

Dear Dr. Sun,

Thank you for submitting your manuscript to PLOS ONE. After careful consideration, we feel that it has merit but does not fully meet PLOS ONE’s publication criteria as it currently stands. Therefore, we invite you to submit a revised version of the manuscript that addresses the points raised during the review process.

We look forward to receiving your revised manuscript.

Kind regards,

Susan Horton

Academic Editor

PLOS ONE

Additional Editor Comments (if provided):

Please see REviewer 1's comments below. I also provided responses as Reviewer 2, which I include here.This paper is somewhat improved in methodology following revisions, although the quality of the written English remains an issue. A big disadvantage is that the authors did not provide a “track changes” version of the manuscript and thus it was hard to see whether the changes really had been implemented as requested. I would request that in the next revision a track changes version be provided.

One issue for me is that “Lowest Priced Generic” was not clearly defined. Does this mean that the authors identified a lowest priced generic brand overall (how?) in the province, and searched for that at the pharmacies? If so, that isn’t that interesting if there are other fairly low-priced generics which are available instead. and it is understandable that that particular brand was not always available. Or in each pharmacy, did they identify the lowest priced generic carried by that particular pharmacy, which wasn’t always available? What is there was another fairly low priced generic available instead? There is a big different in policy implications between no generic alternative being available at all, or another one being available at a slightly higher price.

Abstract: I was puzzled as to why Methods, Findings and Conclusions were repeated twice. Perhaps one was original, one was revised? In any case, under Findings, line 4 (both versions): suggest refer to “innovator brands” instead as “OBs” – confusing to the reader to call the same thing by two different names.

Introduction, para 2: don’t mention the “three in one” policy without a brief explanation – readers will not all be familiar with this. Readers will also not understand what the significance of the “province’s unification” is without explanation.

In many places a space is missing: I will list the words and suggest the authors do a “search and replace”: “anadditional” “mainurban” “theprice” “andcollected” forms.No” “Taizhou,Jiaxing” “Zhejian:one” “ratios(MPRs)” “brands(OBs)” “generics(LPGs)” “work .For” “expenses, [15]and” “studywas”

Table 2: this is a continuation of Table 1, and should have the title “Table 1 (continued)” or similar, and subsequent tables should accordingly be renumbered.

Table 3: suggest add “n=50” to the title for clarity

Table 7: in the column titles, clarify in a footnote perhaps that “Duration” means “duration of treatment course over a one-month period” or similar.

Discussion: second full paragraph discusses “the government’s inadequate subsidies for essential medicines..” – is what is meant that the government does not permit any handling fee for medicines by pharmacies? (That is indeed a problem regarding incentives).

Same paragraph: “.Although hospitals have charged for pharmaceutical services..” (insert underlined word). Also “…pharmaceutical services in Zhejiang or China are in their infancy..” (not “is”)

Paragraph beginning: “The essence of "4 + 7" is to exchange price by quantity.” I don’t understand what the authors mean here. Is what they are saying “The essence of “4+7” is to use larger volumes of purchase to drive down prices”.

“but also has its own policies among various provinces” – not clear what is meant here. Are the authors trying to say “but each province has its own policies”. “The policies of Zhejiang Province encourage public and private pharmacies to increase the availability of medicines and

reduce the affordability of medicines.”. How?

Our research is based on the actual situation of Zhejiang Province, China, and is different from other countries. Other countries do not have the same policies as China, so the research results are also different.” This may be true but is too vague to be useful.

Next page: “Due to confidentiality reasons, some data have errors and missing” – what confidentiality reasons? Please explain

Conclusion: “the government should increase investment” – in what? People don’t usually consider providing a pharmacy service fee for medicines, as investment.

Reviewers' comments:

Reviewer's Responses to Questions

**Comments to the Author**

1. If the authors have adequately addressed your comments raised in a previous round of review and you feel that this manuscript is now acceptable for publication, you may indicate that here to bypass the “Comments to the Author” section, enter your conflict of interest statement in the “Confidential to Editor” section, and submit your "Accept" recommendation.

Reviewer #1: (No Response)

2. Is the manuscript technically sound, and do the data support the conclusions?

Reviewer #1: Yes

3. Has the statistical analysis been performed appropriately and rigorously? 

Reviewer #1: Yes

4. Have the authors made all data underlying the findings in their manuscript fully available?

Reviewer #1: Yes

5. Is the manuscript presented in an intelligible fashion and written in standard English?

Reviewer #1: Yes

6. Review Comments to the Author

Reviewer #1: Dear Authors,

Thank you for the effort towards improving the ms. However, it is very difficult to know if changes were clearly made or points were inserted in the revised version of the ms. Changes were neither highlighted nor indicated by track changes.

Conclusions - Authors are supposed to conclude based on the objectives and main findings. But, the last paragraph, the authors have included a ref i.e. Minghuan Jiang et al, and it looks like a discussion section.

Figure 1 - cetirizine - small letter 'c'. The authors mentioned that this typo had been corrected.

I doubt that proper actions have been taken and please highlighted the corrections and improvements in the revised ms.

7. PLOS authors have the option to publish the peer review history of their article (what does this mean?). If published, this will include your full peer review and any attached files.

Reviewer #1: **Yes: **Mohamed Izham Mohamed Ibrahim

---

## [Author Response · Author response to Decision Letter 1]

20 Aug 2020

Dear Susan Horton

Thanks you very much for your comments and suggestions. 

As suggested, We have made amendments to the academic papers, in accordance with the requirements of the academic editors, and responded to the comments made by the reviewers. The details are attached below. Some of our basic research data is written in Chinese, which may be difficult to understand. Our research data has been uploaded to the submission system. Because it is the most primitive data, it is all in Chinese。

We have revised the manuscript, according to the comments and suggestions of reviewers and editor, and responded, point by point to, the comments as listed below. Some of the references we cited are published in domestic academic journals in China and may not be found in foreign databases.

I would like to re-submit this revised manuscript to PLOS ONE, and hope it is acceptable for publication in the journal.

Looking forward to hearing from you soon.

With kindest regards.

Yours Sincerely

Guojun Sun

---

## [Editor Report · Decision Letter 2]

27 Aug 2020

PONE-D-19-30034R2

Availability, prices and affordability of essential medicines in Zhejiang Province, 

China

PLOS ONE

Dear Dr. Sun,

Thank you for submitting your manuscript to PLOS ONE. After careful consideration, we feel that it has merit but does not fully meet PLOS ONE’s publication criteria as it currently stands. Therefore, we invite you to submit a revised version of the manuscript that addresses the points raised during the review process.

We look forward to receiving your revised manuscript.

Kind regards,

Susan Horton

Academic Editor

PLOS ONE

Additional Editor Comments (if provided):

Thank you for responding to the previous comments. There is one outstanding issue. "Lowest priced generic" is still not well-defined. Is this one particular brand which is deemed the lowest-priced in the province? (this is what it seems to be)? If not, please define out of which subset it is "lowest priced".

Thank you for explaining the policy in the conclusions more fully. Unfortunately the language of the draft is unacceptable at the moment. At this point you need to pay someone for whom English is their first language and who has some understanding of the subject matter to edit the draft. This will be the third revision of this article. I strongly recommend that you perfect the language this time since I do not know if the journal will accept further revisions (this is up to the editor).

---

## [Author Response · Author response to Decision Letter 2]

6 Oct 2020

Dear Susan Horton

Thanks you very much for your comments and suggestions. 

As suggested, We have made amendments to the academic papers, in accordance with the requirements of the academic editors, and responded to the comments made by the reviewers. The details are attached below. We have asked editge to help us polish the language and correct the previous language problem.

We have revised the manuscript, according to the comments and suggestions of reviewers and editor, and responded, point by point to, the comments as listed below. Some of the references we cited are published in domestic academic journals in China and may not be found in foreign databases.

I would like to re-submit this revised manuscript to PLOS ONE, and hope it is acceptable for publication in the journal.

Looking forward to hearing from you soon.

With kindest regards.

Yours Sincerely

Guojun Sun

---

## [Editor Report · Decision Letter 3]

12 Oct 2020

PONE-D-19-30034R3

Availability, prices and affordability of essential medicines in Zhejiang Province, 

China

PLOS ONE

Dear Dr. Sun,

Thank you for submitting your manuscript to PLOS ONE. After careful consideration, we feel that it has merit but does not fully meet PLOS ONE’s publication criteria as it currently stands. Therefore, we invite you to submit a revised version of the manuscript that addresses the points raised during the review process.

We look forward to receiving your revised manuscript.

Kind regards,

Susan Horton

Academic Editor

PLOS ONE

Additional Editor Comments (if provided):

Thank you for getting help with the English. Now that I can understand better what you are trying to say, I can now offer the following edits so that the readers of the journal will also understand. I recommend the following edits. Since there do not seem to be page or line numbers, I have numbered according to the page number in the proof which overall has 53 pages.

P13 of 53: “LPGs are the cheapest generic drugs found in the research institute” Do you mean “LPGs are the cheapest generic drugs found by the University research institute”?

p13 pf 53 “Data collection included six cities in Zhejiang Province. Hangzhou was identified as the major urban center of Zhejiang, and an additional five cities were chosen randomly. As a result, six cities were selected: Hangzhou, Ningbo, Shaoxing, Huzhou, Taizhou, and Jiaxing”. Delete this sentence – duplicates statement previously. In following sentence say “area surveyed” not “surveyed area”

In several places the authors say “We refer to Table X”. This is unnecessary, as long as the table is cited in the text.

P17 of 53, last 2 lines (last 2 lines of Table 2). These should be deleted.

P19 of 53. Restate the following sentence: ,”The results in Table 5 show that the patient price in the public sector is 13.41 lower than that in the private sector for OBs, while it is 5.21 higher for LPGs.” As “,The results in Table 5 show that the patient price in the public sector is 13.41 for OBs, lower than that in the private sector while for LPGs it is 5.21, higher than that in the private sector “.

P20 of 53: “treating peptic ulcer with omeprazole required 5.2 days’ wages.” Treating for how long (the recommended course is 4-8 weeks).

P21, last line. “the mean availability of OBs and LPGs was 12.9% and 29.5%, respectively.[11].” Is this in private pharmacies? If so, state that.

P22 of 53, last line. Does “social” pharmacies mean “public”? If so, make this change from here onwards (used several times).

P23 of 53: I think instead of “4+7 belt quantity purchasing policy” what is means is “4+7” bulk purchasing policy. State that.

P23 of 53, last line. If the average price of medicines was reduced by more than 60%, what exactly was reduced by 92%?

P24 of 53, line 4: instead of “burden of patients” say “burden on patients”.

P24. Delete entire paragraph (duplicates material elsewhere “The essence of “4+7” policy is to use larger purchase volumes to drive down prices. Successful bidders who pass the conformity evaluation of generic drugs will

greatly reduce the price of drugs. The government will give the winning bidder a

certain market share. The coverage of high-quality and low-price generic drugs will

expand; the pressure to make drugs affordable will be reduced; and patients will have

better drug choice.[19Some pharmaceutical industry associations apply for

implementation of the policy to be suspended, but the decision of the health insurance

department to implement the policy is very resolute. Because of their own interests,

pharmaceutical enterprises will strictly control the quantity of drugs produced,

resulting in a rapid reduction in the availability of drugs.[20]”

p24. Also delete sentence “In China, each province has its own policies, and

the provincial government will respond to the official documents of the Chinese

government and issue relevant policies based on the development of its own province.” Since this simply restates what was said in the previous sentence.

P25: delete “This reflects the examination of the self-procurement system of public

medical institutions. In this study, we adopted a systematic research method to study related issues, combined with the actual situation and policies to study the availability and

affordability of drugs in public and private pharmacies in Zhejiang”. This simply restates what has been previously said.

P26: delete “The availability and affordability of medicines is a complex systemic problem.

The question first needs to clarify the essential causes of the shortage of drugs; on this

basis, grasp the key factors; and propose solutions in regard to drug prices, market

environment, policy systems, and legislation. The smooth implementation of these

solutions also requires the coordination and cooperation of various government

departments and stakeholders of the drug supply. Only when the main bodies of the

drug supply chain work together can the current issues of great concern be solved to

the greatest extent.” Also repetitious.

P26: Delete “In terms of the development trend,” (repetitious)

P26: what is meant by “the income of the basic medical insurance fund”? Does this mean aggregate income of the fund in the province, or does it mean “each worker’s basic medical insurance allotment”

P26: “Second, the government should further play the role of social

pharmacies in improving the availability and affordability of essential medicines”. Suggest restate as “Second, the government should improve the availability and affordability of essential medicines in public pharmacies”.

P27. Delete “with diagnosis and treatment of behavior belonging to shall, through the

proper way to manifest the value of their labor”: sentence is clearer without this.

---

## [Author Response · Author response to Decision Letter 3]

20 Oct 2020

Dear Susan Horton

Thanks you very much for your comments and suggestions. 

As suggested, We have made amendments to the academic papers, in accordance with the requirements of the academic editors, and responded to the comments made by the reviewers. We have made changes based on the previous manuscript.The details are attached below. We have asked editge to help us polish the language and correct the previous language problem.

We have revised the manuscript, according to the comments and suggestions of reviewers and editor, and responded, point by point to, the comments as listed below. 

I would like to re-submit this revised manuscript to PLOS ONE, and hope it is acceptable for publication in the journal.

Looking forward to hearing from you soon.

With kindest regards.

Yours Sincerely

Guojun Sun

---

## [Editor Report · Decision Letter 4]

21 Oct 2020

Availability, prices and affordability of essential medicines in Zhejiang Province, 

China

PONE-D-19-30034R4

Dear Dr. Sun,

We’re pleased to inform you that your manuscript has been judged scientifically suitable for publication and will be formally accepted for publication once it meets all outstanding technical requirements.

Kind regards,

Susan Horton

Academic Editor

PLOS ONE
---

## [Editor Report · Acceptance letter]

5 Nov 2020

PONE-D-19-30034R4 

Availability, prices and affordability of essential medicines in Zhejiang Province, China 

Dear Dr. Sun:

I'm pleased to inform you that your manuscript has been deemed suitable for publication in PLOS ONE. Congratulations! Your manuscript is now with our production department. 

Kind regards, 

on behalf of

Dr. Susan Horton 

Academic Editor

PLOS ONE